# Physiologically Based Pharmacokinetic Model of CYP2D6 Associated Interaction Between Venlafaxine and Strong Inhibitor Bupropion—The Influence of Age-Relevant Changes and Inhibitory Dose to Classify Therapeutical Success and Harm

**DOI:** 10.3390/pharmaceutics17020179

**Published:** 2025-01-31

**Authors:** Ulrich Ruben Luecht, Wolfgang Scholz, Ann-Kathrin Geiben, Ekkehard Haen, Georg Hempel

**Affiliations:** 1Department of Clinical Pharmacy, Institute of Pharmaceutical and Medical Chemistry, University of Muenster, Corrensstrasse 48, 48149 Muenster, Germany; ulrich.luecht@uni-muenster.de; 2ePrax GmbH, Dessauerstrasse 9, 80992 Munich, Germany; 3Clinical Pharmacology, Department of Pharmacology and Toxicology and for Psychatry and Psychotherapy, University of Regensburg, Universitätsstrasse 84, 93053 Regensburg, Germany; ann-kathrin.geiben@klinik.uni-regensburg.de (A.-K.G.);

**Keywords:** PBPK, drug–drug interaction, therapeutic drug monitoring, multiple drug–drug interaction (MDDI), depression therapy, venlafaxine, bupropion

## Abstract

**Background/Objectives:** Venlafaxine (VEN) is commonly used in young and elderly patients. Bupropion (BUP) is occasionally added to depression treatments with VEN. BUP’s inhibitory potential toward CYP2D6, VEN’s main metabolic pathway, may provoke a higher risk for toxic or adverse drug effects. Therefore, the question arises if a dose reduction in VEN or BUP is needed to avoid clinically relevant changes in exposure to VEN and its metabolite O-desmethylvenlafaxine (ODV). **Methods:** The literature-based PBPK models of VEN, BUP and their active metabolites under single-dose and steady-state conditions were created by using PK-Sim^®^. To evaluate the DDI model‘s predictive performance, trough plasma concentrations (<65 years, *n* = 54 and ≥65 years, *n* = 13) of VEN/ODV were extracted from the TDM database KONBEST. DDI’s clinical extent was assessed by AUC changes in VEN, ODV and active moiety (AM). The prediction was compared to the results of SCHOLZ Databank’s MDDI calculator (MDDIcalc). **Results:** Models accurately describe VEN’s and BUP’s pharmacokinetics and BUP’s effect on VEN’s metabolism in the age strata. The model predicts higher exposure to VEN (+110% to 132%), lower exposure to ODV (−50.0% to −61.5%) and a negligible change in AM (−1.02% to −2.40%). The AUC changes increase with higher BUP doses but is independent of patients’ age. Because of the missing AUC change in the AM, the DDI is considered clinically irrelevant. The MDDIcalc predicts no relevant effect on the AUC of AM with BUP. **Conclusions:** Both PBPK and MDDIcalc provide, in their own way, valuable tools to predict the DDI’s extent. Further research is needed regarding elderly patients, renal or hepatic impairment and polymorphisms, especially CYP2D6, CYP2C9, CYP2C19 and UGT.

## 1. Introduction

Administration of more than one drug is ubiquitous in pharmacotherapy, especially in the treatment of depression and mental illnesses due to multimorbidity, with higher risks for elderly patients caused by polypharmacy [1,2]. With each additional drug in the medication regimen, the incidence of pharmacokinetic and pharmacodynamic drug–drug interactions (DDIs) [3] and the risk of harming patients’ safety because of potentially fatal adverse drug reactions (ADRs) increase or therapeutic success decreases because of the loss of efficacy or non-adherence by patients as a result of increasing ADRs. In practice, the prescription of drug combinations like venlafaxine (VEN) and bupropion (BUP) is common. The individuality in patients’ pharmacokinetics requires a constant re-evaluation of the concomitant use of DDI-causing drugs and quantifying their extent by Therapeutic Drug Monitoring (TDM) and the creation of physiology-based pharmacokinetic (PBPK) models with the aim of making drug therapy safe, tolerable and harmless [4]. It may be required to reduce the dose of the inhibitory drug or the substrate. Likewise, discontinuation or avoidance of the combination may be an option for this purpose. The aim of developing this PBPK model is its application in routine clinical practice; patients can be generated based on their demographics, and the resulting plasma concentrations can be simulated. This allows the attending physician to determine whether combined administration is possible before the drug is administered, thus contributing to personalized medicine.

### 1.1. Venlafaxine

VEN is a norepinephrine and serotonin reuptake inhibitor and is indicated for the treatment of major depression, generalized or social anxiety and panic disorder with or without agoraphobia. VEN is marketed in extended-release and immediate-release dosage forms. The immediate-release form is used much less frequently; this is because the extended-release form has greater patient adherence, as it is administered only once daily, has a lower risk of ADRs like nausea and provides better therapeutic efficacy [5]. For this reason, the focus of this study is on VEN’s extended-release dosage form. Its active metabolite, O-desmethylvenlafaxine (ODV), has similar pharmacological effects and therapeutic efficacy. A steady state is established within three days of steady daily intake. The bioavailability is not affected by the administration of food or the timing of intake. VEN undergoes extensive hepatic metabolism mainly by CYP2D6 to its active metabolite ODV [6]. The area under the curve (AUC) of ODV is two-fold higher than the AUC of the parent drug at a steady state. ODV is formed particularly by CYP2D6 and, to a lesser extent, by CYP2C19 and CYP2C9 catalysis, and, afterward, it is glucuronidated, to a large extent, by UDP-glucuronyltransferase (UGT) catalysis and, to a very small extent, N-desalkylated by CYP3A4 to N,O-didesmethylvenlafaxine (NODV). CYP2D6 is not involved in the metabolism of ODV [7]. Other metabolic pathways of VEN are via CYP3A4 [8]. According to the prescribing information [6], no dose adjustment is necessary in elderly patients. However, the lowest effective dose should be used. According to the “Consensus Guideline for Therapeutic Drug Monitoring in Neuropsychopharmacology” by Hiemke et al., the therapeutic reference range, related to the active moiety (AM, sum of the plasma concentration of VEN and ODV) is between 100 and 400 ng/mL. The toxic threshold level is 800 ng/mL in plasma for AM [9].

### 1.2. Bupropion

BUP is a norepinephrine and dopamine reuptake inhibitor indicated for the treatment of episodes of major depression and is marketed only with extended-release dosage forms in Germany. BUP is metabolized in an extensive, stereoselective way in humans. Among others, three active metabolites—hydroxybupropion (OHB, potency is 50% of BUP), threohydrobupropion (THB, potency is 20% of BUP) and erythrohydrobupropion (EHB, potency is like THB)—are formed. OHB, THB and EHB are further metabolized to inactive metabolites [10,11]. A key enzyme in BUP metabolism is CYP2B6, which catalyzes hydroxylation to OHB, which is present in the largest proportion of plasma (which has an AUC of about 13-fold higher than BUP). CYP1A2, 2A6, 2C9, 2C19, 3A4 and 2E1 play subordinate roles. The reduction in BUP to THB and EHB proceeds under catalysis of 11β-hydroxysteroid dehydrogenase 1 (HSD) [10]. After metabolization via CYP or HSD, OHB, THB and EHB undergo UGT-catalyzed glucuronidation [12] and THB and EHB hydroxylation under catalysis of CYP2C19 [13]. The AUC of EHB in steady state is similar to its parent drug, and the AUC of THB is about seven-fold higher than BUP. Unchanged excretion of BUP accounts for only about 0.5% of the dose; the proportion of other metabolites in urine is about 10%, with unchanged THB accounting for the largest proportion of about 6%. Due to the risk of BUP and its metabolites accumulating as a result of declining kidney function in older age, a maximum dose of 150 mg per day should be used in elderly patients [10]. Concurrent ingestion with food showed no effect on the AUC and C_max_. Steady-state plasma concentrations of BUP, OHB, THB and EHB are reached within about eight days [10]. BUP, OHB, THB and EHB are capable of inhibiting CYP2D6 without being metabolized [14]. Therefore, a concomitant administration of BUP and CYP2D6 substrates like VEN should be performed with caution.

The aim of this work was to show the extent to which the combination with BUP affects the pharmacokinetics of VEN and if dose adjustment to minimize the risk of ADRs makes sense when considering different age groups of patients by using PBPK modeling of both VEN and BUP. Literature-based models for single and multiple administration were created and their results were finally combined in a DDI approach. We choose PBPK instead of population pharmacokinetics (PopPK) because PbPk allows a more mechanistic insight into pharmacokinetics. To our knowledge, this is the first PBPK model conducted with PK-Sim^®^ for the DDI between VEN and BUP using TDM data and including active metabolites and AM. We chose the interaction of VEN and BUP because they are occasionally administered together and the metabolization of VEN occurs primarily via CYP2D6. BUP was chosen as a strong CYP2D6 inhibitor because the combination of other strong inhibitors of the enzyme such as fluoxetine and paroxetine is not used therapeutically. This was also shown by the analysis of the TDM data from the KONBEST database used for the evaluation of the model. Elderly patients often take multiple medications (polymedication) and are frequently excluded from studies. To our knowledge, no pharmacokinetic studies have yet been conducted on the interaction under investigation. PBPK modeling is therefore well suited to explore this interaction in more detail, particularly concerning age-related changes, without posing risks to patients. As interaction databases are also becoming increasingly popular, a corresponding tool in the form of the Multi-Drug–Drug Interaction (MDDI) calculator (MDDIcalc) as part of the SCHOLZ Databank (SDB) was also included in this study for comparison purposes.

## 2. Materials and Methods

### 2.1. Physiologically Based Pharmacokinetics (PBPK) Modeling

Modeling was performed using PK-Sim^®^ version 9.1 and MoBi^®^ version 9.1 as part of the Open Systems Pharmacology Suite (download: http://www.open-systems-pharmacology.org/ (accessed on 30 December 2024)). The software (version 9.1) implements a generic structure consisting of 17 different compartments, each representing a tissue or organ. All compartments are connected to arterial and venous blood pools by organ-specific blood flows to allow intercompartmental transport of substances. Each organ is subdivided into subcompartments, e.g., vascular space containing erythrocytes and plasma, intercellular space and interstitial tissue. Based on the physicochemical properties (e.g., plasma protein binding or fraction unbound, lipophilicity and molecular weight), compound-specific clearance information and organ/tissue composition data (e.g., lipids, proteins, water), PK-Sim^®^ estimates drug-associated models. For the description of the passive physical processes involved in partitioning into the organs, blood flow-limited partitioning was used. For a more detailed overview, refer to the Open Systems Pharmacology Suite manual [15].

With the support of PK-Sim^®^, it is possible to quantitatively predict the extent of DDI. In this way, comedications influencing pharmacokinetics through inhibitory or inducing efficacy can be identified in the development of new drugs before they are administered to patients, or the extent to which the new substance itself influences other drugs can be analyzed. The possibility of using PBPK-DDI approaches is recommended by the Food Drug Agency (FDA) and European Medicines Agency (EMA) to provide dosing recommendations for the comedication of interacting drugs or the use of combinations in special populations in clinical trials [4,16].

#### 2.1.1. General Workflow

Based on a systematic literature survey to detect relevant anatomical, physicochemical and drug-specific parameters like absorption, distribution, metabolism and elimination of VEN, ODV, as well as BUP, OHB, THB and EHB, an initial multiple dose model was built for both antidepressants. The models were evaluated by using study data, extracted from seven (VEN [17,18,19,20,21,22,23]) and six (BUP [24,25,26,27,28,29]) pharmacokinetic studies, respectively (more details in the Appendix A). Both developed models were finally combined in a DDI approach. Plasma trough concentrations (C_tr_) of VEN and ODV extracted from the database KONBEST were used to evaluate the predictability of the DDI model. KONBEST is a large TDM database of the University Hospital Regensburg. Plasma concentrations of psychotropic drugs have been recorded [30] based on the information published in the TDM Consensus Guidelines of the Arbeitsgemeinschaft für Neuopsychopharmakologie und Pharmakopsychatrier (AGNP) [9] when determining the concentrations of parent substances and their active metabolites, if present. The used data were recorded in the database from 2005 to 2018 and were collected in various institutions of the Arbeitsgemeinschaft Arzneimitteltherapie bei psychischen Erkrankungen (AGATE) and at the University Hospital of the Rheinisch-Westfälische Technische Hochschule Aachen. The data were transmitted in anonymized form without the possibility of tracing personal data such as date of birth, name and address of the patients and was evaluated only retrospectively. The KONBEST database has not been updated since 2019. The change in the AUC of VEN, ODV and AM was used to evaluate the clinical relevance of the DDI. The concentration of AM was integrated into PK-Sim^®^ via MoBi^®^ by using Equation (1), which is a modification of the equation published by Kneller et al. [31]. In this equation, art is arterial, bon is bone, c is concentration in plasma, fQ is blood flow rate, M_r_ is relative molecular weight, mus is muscle and skn is skin.(1)$$c_{AM}={\left(\left(\begin{array}{c} {fQ}_{art}\times c_{art}+{fQ}_{bon}\times c_{bon}+{fQ}_{fat}\times c_{fat}+\\ {fQ}_{mus}\times c_{mus}+{fQ}_{skn}\times c_{skn} \end{array}\right)\times M_{r}\right)}_{VEN}+\;{\left(\left(\begin{array}{c} {fQ}_{art}\times c_{art}+{fQ}_{bon}\times c_{bon}+{fQ}_{fat}\times c_{fat}+\\ {fQ}_{mus}\times c_{mus}+{fQ}_{skn}\times c_{skn} \end{array}\right)\times M_{r}\right)}_{ODV}$$

#### 2.1.2. DDI Implementation

The process of CYP2D6 inhibition by BUP and its metabolites is integrated as competitive in the model, as described by Reese et al. [14]. Using parameter identification, the values of inhibitory constant K_i_ from the literature were approximated to the concentration–time courses of VEN and ODV with simultaneous administration of BUP, considering the strength of the inhibitory potential on CYP2D6 measured in vitro (Table 1). For the DDI model approach, CYP2D6 polymorphisms were not considered, because they were not investigated and documented by KONBEST.

### 2.2. Development of Literature-Based PBPK Model

For both, VEN and BUP literature-based models were built. For simulating the different studies, subjects‘ demographics were used to create an output individual based on the published mean and a population of *n* = 1000 individuals, based on the published ranges of age, weight and height (demographics can be found in the Appendix A). If there was a lack of demographic data, subjects were characterized based on a standard European (male, 30 years, 73 kg, 176 cm and a BMI of 23.57 kg/m^2^) in the age range of 20 to 50 years. Plasma concentrations of VEN, ODV, BUP, OHB, EHB and THB were extracted by using WebPlotDigitizer Version 4.4 (Ankit Rohatgi, Pacifica, CA, USA).

Drug- and metabolite-related physicochemical data and parameters to describe the absorption, distribution, metabolism and excretion were obtained from the literature. Metabolizing enzymes were integrated into the models and renal clearance values were added. High variability in the pharmacokinetics of VEN and BUP are considered in population simulations by implementing postulated standard deviations of V_max_, K_M_ and renal clearances and minimal to maximal ranges, respectively. For VEN, the catalytic rate constant (k_cat_), also known as the turnover frequency for CYP2D6, CYP2C9, CYP2C19 and CYP3A4, as well as intrinsic clearance of UGT for glucuronidation of ODV, were fitted as close as possible to the mean plasma concentrations with the aid of parameter optimization. For CYP2D6 poor metabolizers, k_cat_ values were set to 0 min^−1^ in the representation of no metabolizing capacity [32]. The same method was performed during the BUP model investigation for involved metabolizing processes. Input parameters can be found in Table 2. Metabolism and inhibition are shown schematically in Figure 1.

VEN and BUP are commercially available as extended-release tablets [6,10]. To model the release of the active ingredients from the oral formulations, parameter identification of the 50% dissolution time t, lag time t_lag_ and shape parameter b of the Weibull function integrated into PK-Sim^®^ was carried out using a Monte Carlo algorithm with the default settings of PK-Sim^®^. It was considered that the calculated values approximate the C_max_ described in the literature at the time of t_max_.

The model evaluation and adaptation steps were carried out in accordance with the EMA guideline [33]. The in vivo data described in the literature composed of the measured plasma concentration–time courses of the parent substances and their metabolites, and the PK parameters C_max_, AUC, t_max_ and the percentage of drug excreted in the urine, were used for continuous model evaluation, by calculating prediction errors (PE, Equation (2)). The mean prediction error (MPE, bias, Equation (3)) and mean absolute prediction error (MAPE, precision, Equation (4)) of all extracted plasma concentrations were determined to check the model accuracy and for illustration, goodness-of-fit (GOF) plots were created. The results of literature-based modeling and their evaluation can be found in the Appendix A.(2)$$PE\;\left[\%\right]=\;\frac{predicted-observed}{observed}\;\times 100$$(3)$$MPE\;\left[\%\right]=\;\frac{1}{n}\;\times \;\sum_{i=1}^{n}{PE}_{i}$$(4)$$MAPE\;\left[\%\right]=\;\frac{1}{n}\;\times \;\sum_{i=1}^{n}\left|{PE}_{i}\right|$$

**Table 2 pharmaceutics-17-00179-t002:** Input parameters for PBPK models of VEN and BUP.

Parameter	VEN	ODV	Metabolism Parameter VEN → ODV			
M_r_ [g/mol]	277.4 [34]	263.38 [35]	CYP	V_max_	K_M_	k_cat_
logP	2.62 [2.8 [34], 2.69, 2.74 [6]]	2.45 [2.62 [35], 2.6, 2.29 [36]]	2D6 n.d. 2D6 EM 2D6 PM	6.48 [7]	23.2 [7]	29.16 * 64.8 * 0 *
pKa	9.4 [34]	8.86 [35], 10.25 [35]	2C9	0.58 [7]	3119 [7]	2.61 *
Solubility	572 [6]	-	2C19	3.78 [7]	293 [7]	17.01 *
f_u_ [%]	73 [6]	70 [6]	VEN → Sink			
Part. Coef.	Berezhkovskiy [37]	Berezhkovskiy [37]	CYP	V_max_	K_M_	k_cat_
Cell. Perm.	PK-Sim Standard	PK-Sim Standard	3A4	1.23 [7]	556 [7]	5.54 *
T_diss_	338 *		2C9	10.33 [7]	2250 [7]	46.49 *
T_lag_	81.2 *		2C19	7.56 [7]	398 [7]	34.02 *
Diss. Sh.	0.92 *		ODV → Sink			
Cl_renal_	0.05 ± 0.02 [38]	0.12 ± 0.03 [39]		Cl_int_ [min^−1^]		
CYP_2D6_ K_i_	41 ± 9.5 [40]	40 [41]	UGT_1A1_	0.1 *		
parameter	BUP	OHB	EHB		THB	
M_r_ [g/mol]	239.74 [36]	255.74 [36]	241.76 [42]			
logP	3.5 [3.6 [36], 3.2 [43], 3.5 [44]]	1.98 [2.6 [44], 1.98 [36], 2.03 [42]]	2.69 [2.88 [42], 2.69 [45]]			
pKa	7.9 [7.9 [46], 8.0 [43], 7.2 [44]]	7.7 [44]	9.6 [42]			
Solubility	312 [47]	-	-			
f_u_ [%]	16 [14]	23 [14]	58 [14]			
Part. Coef.	Schmitt [48]	Rodgers + Rowland [49,50]	Berezhkovskiy [37]			
Cell. Perm.	PK-Sim Standard	PK-Sim Standard	Charge-dependent Schmitt			
T_diss_	170 *					
Diss. Sh.	1.52 *					
Cl_renal_	0.17 (0.12–0.21) [27]	0.02 (0.02–0.03) [27]	0.50 (0.39–0.61) [27]		0.36 (0.28–0.45) [27]	
CYP_2D6_ K_i_	0.46 * [21 [14]]	0.41 [13.3 [14]]	0.15 [5.4 [14]]		0.04 [1.7 [14]]	
	CYP_2B6_ BUP → OHB	UGT_2B7_ OHB → Sink	CYP_2C19_ THB → Sink		CYP_2C19_ EHB → Sk.	
V_max_	3623 ± 1520 [51] ^†^	5550 ± 507 [12] ^†^	0.55 [13]		0.40 [13]	
K_M_	89 ± 14 [51]	488 ± 98.3 [12]	13.0 [13]		39.0 [13]	
k_cat_	92.9 *	33.81 *	1.10 *		0.80 *	
	HSD_11ß1_ BUP → THB	UGT_2B7_ OHB → Sk.	UGT_1A9_ THB → Sink		UGT_2B7_ EHB → Sk.	
V_max_	11,800 ± 265 [52] ^†^	739 ± 59.6 [12] ^†^	3290 ± 269 [12] ^†^		2280 ± 241 [12] ^†^	
K_M_	42.2 ± 3.05 [52]	172 ± 38.9 [12]	343 ± 37.5 [12]		360 ± 55.2 [12]	
k_cat_	102.7 *	5.27 *	38.1 *		52.8 *	
	HSD_11ß1_ BUP → EHB		UGT_2B7_ THB → Sink		UGT_2B7_ EHB → Sk.	
V_max_	661 ± 54 [52] ^†^		358 ± 25.5 [12] ^†^		1740 ± 212 [12] ^†^	
K_M_	66.5 ± 19.9 [52]		248 ± 27.1 [12]		373 ± 63.0 [12]	
k_cat_	38.46 *		4.14 *		40.3 *	

BUP: bupropion, Cell. Perm: cellular permeability, Cl_int_: intrinsic clearance, CYP: cytochrome P450, Diss. Sh.: dissolution shape, EHB: erythrohydrobupropion, EM: extensive metabolizer, f_u_: fraction unbound to plasma, HSD: hydroxysteroiddehydrogenase, k_cat_ [min^−1^]: turnover frequency, K_i_ [µM]: inhibitory constant, KM [µM]: Michaelis–Menten constant, logP: logarithm of octanol/water partition coefficient, M_r_ [g/mol]: molecular weight, n.d.: not defined, ODV: O-desmethylvenlafaxine, OHB: hydroxybupropion, Part. Coef.: partition coefficient, pKa: negative decadic logarithm of acid dissociation constant, PM: poor metabolizer, Sink/Sk.: not defined metabolite(s), solubility in water at pH = 7 [mg/mL], t_diss_ [min]: time to dissolve 50%, t_lag_ [min]: lag time, THB: threohydrobupropion, UGT: uridinglucoronosyltransferase, VEN: venlafaxine, V_max_ [pmol/min/pmol]: maximal velocity of the reaction. * calculated and fitted by PK-Sim^®^. ^†^ change in unit in pmol/min/mg protein.

### 2.3. Development of DDI Model

To characterize the simulated population of *n* = 1000 in PK-Sim^®^, respective medians of the age strata of patients extracted from KONBEST were used as the demographic data of the initial individual and the minimum and maximum values were used as limits for the population representation (Table 3).

The KONBEST-DDI dataset includes patients receiving VEN and BUP in combination, but no other drugs possibly affecting the metabolism of VEN. Patients receiving other strong CYP2D6 inhibitors like paroxetine, fluoxetine and cinacalcet and the proton pump inhibitors (es)omeprazole and pantoprazole, which inhibit CYP2C19, were excluded. Other exclusion criteria were missing demographic data of the patients, missing concentrations of VEN and/or ODV, dosages and doses that indicate immediate release formulations, changes in dosage or the starting of therapy less than three days before blood sampling and documentation bias.

The dataset was stratified by age into young (18–64 years) and older (≥65 years) patients. The extracted C_tr_ of VEN and ODV (*n* = 57 and *n* = 13 for elderly patients) were used for verifying the DDI model predictive performance. The combination of 225 mg VEN and 150 or 300 mg BUP, respectively, was simulated in both age groups. Dose normalization was performed to harmonize all VEN doses that were combined with BUP. The dosing regimen of the PBPK model was structured as follows: First, individuals created by PK-Sim^®^ received 225 mg VEN once daily for four days. On the fifth day, the additional administration of BUP was initiated once daily, administered at the same time as the VEN dose. The combination of the two psychotropic drugs was simulated over a total of 14 days. The BUP administration was then discontinued.

#### 2.3.1. DDI Model Evaluation

The predictive power of the interaction model was evaluated using C_tr_ of VEN, ODV and AM from KONBEST. The time of C_tr_ was defined as the difference between the last change in BUP administration (initiation or dose adjustment of BUP) and the day of sampling and was placed immediately before the subsequent dose. If the last change was longer than ten days ago, the sample date was assumed as day 10 of concomitant use, because then BUP and its active metabolites have reached their steady-state conditions. Extrapolation to older patients was performed under the same conditions as described above. Due to the small number of patients, one population was used here for each age group with both BUP doses (Table 3).

To assess the effect of concomitant administration of BUP, the AUC change in VEN, ODV and AM was determined. For this purpose, the simulated AUC of the median of the fourth day was compared with the simulated AUC of the median of day 18 according to Equation (2). The simulated AUCs were also compared according to Equation (5) in order to calculate the change in the metabolic ratio of the exposure MR_AUC_. The deviation of these before and after inhibition was calculated according to Equation (2), as previously described.(5)$${MR}_{AUC}=\;\frac{{AUC}_{ODV}}{{AUC}_{VEN}}$$

To evaluate the clinical relevance of the DDI, the location of the median C_tr_ of AM before and during concomitant use of BUP was observed. If the median C_tr_ of AM is above the therapeutic reference range (100 to 400 ng/mL) or the toxic level (>800 ng/mL) during concomitant use, a dose adjustment of the perpetrator or victim drug may be needed [9].

#### 2.3.2. Excursus: Multiple Interaction of VEN, BUP and Itraconazole (ITRA)

Finally, a possible scenario was simulated in which the combination of 300 mg/d BUP and 225 mg/d VEN was supplemented with a twice-daily (every twelve hours) dose of 100 mg ITRA over seven days. The PK-Sim^®^ template of ITRA and its active metabolite hydroxyitraconazole was used for this purpose. Since ITRA is a strong CYP3A4 inhibitor and VEN is also metabolized via this enzyme, this excursus offers an interesting insight into the effect of polymedication if the two main metabolic pathways of VEN are inhibited. This enables us to answer the question of whether such a combination would necessitate a dose adjustment without endangering patients.

The simulated population was based on a standard European (male, 30 years, 73 kg, 176 cm, 23.57 kg/m^2^). The young population was between 18 and 64 years old while the older population consisted of virtual patients between 65 and 100 years old. A total of 1000 test persons were simulated with a 50% female proportion. The administration protocol for the MDDI simulation contained 6 days of 225 mg VEN use alone, followed by 14 days of VEN +300 mg BUP, followed by 7 days of VEN + BUP + 100 mg ITRA twice daily. To assess the effect of the DDI, the C_tr_ of VEN, ODV and AM was compared on day 4 (without BUP and ITRA), day 17 (with BUP but without ITRA) and day 27 (with BUP and ITRA). For comparison, the twice-daily administration of 100 mg ITRA without concomitant BUP comedication was also simulated and the effect on the AUC of VEN, ODV and AM was investigated.

### 2.4. SCHOLZ Databank’s MDDI Calculator

SDB’s MDDIcalc integrates the calculation of the quantitative effect of multiple drug interactions in the context of polypharmacy on the AUC and derives a recommendation for a possible dose adjustment. MDDIcalc is based on a model of scaled drug properties having an impact on the velocity-determining kinetic processes of the transport, metabolism and elimination of drugs [53]. The MDDIcalc considers the changes in the concentration of all drugs as linear processes (first-order kinetics). The calculations of the relative change in AUC and dosage are independent of the individual dosage. Inhibition of enzymes and transporters can be calculated, while enzyme induction cannot be assessed. If there was no selection of genetic polymorphisms (including PM and intermediate metabolizers (IM) of CYP2D6, CYP2C19 and the organic anion transporter (OATP1B1)), the MDDIcalc computes the change in AUC with the assumption of extensive metabolizers. Prodrugs are taken into account in different ways: alerts are displayed if enzyme inhibition might cause inactivation of the active principle of the drug; if enzyme inhibition modulates the active ingredient and the active metabolite and does not affect the efficacy of the drug, the drug exposure assessed includes the impact of potentially active metabolites. Furthermore, the stages of renal insufficiency are also included in the calculation [54].

The exact functionality of the MDDIcalc is a trade secret and is therefore not disclosed. A publication is in preparation that will provide a more detailed insight into how the MDDIcalc works. To compare the results of the PBPK model, the combinations of VEN/BUP, VEN/BUP/ITRA and VEN/ITRA were entered into the MDDIcalc, and the respective AUC change in AM (VEN subsuming ODV) was extracted.

## 3. Results

### 3.1. Literature-Based Model of VEN and BUP

The literature used, the plasma concentration–time curves and an exact evaluation of literature-based models of VEN/ODV, as well as BUP/OHB/THB/EHB, can be found in the Appendix A. Single- and multiple-dose PBPK models accurately describe plasma concentrations of VEN and ODV in the study populations extracted from the literature. The used formulation of extended-release VEN was well described with a mean t_max_ of 6.8 ± 0.5 h (range of PE = 3.85–26.9%), which follows the prescribing information of the manufacturer [6].

Metabolism of VEN to ODV and sink metabolites proceeds extensively—just a small amount of unchanged VEN was found in the model’s integrated urine container (5.40 ± 0.06%, PE = 14.9%). VEN and ODV pharmacokinetics were predicted precisely with a low bias (MPE) and a good mean precision (MAPE) < 35% (<20% under steady-state conditions). The GOF plot (Figure 2) illustrates, remarkably, that 58.4% (steady state of 91.8%) of all plasma concentrations are in a 1.25-fold and 89.4% (steady state of 100%) are in a twofold error range. Accurate PE in C_max_ (PE_VEN_ −17.0% to 45.6%, Ø 6.15%; PE_ODV_ −38.0% to 26.9%, Ø −10.9%) and AUC (PE_VEN_ −15.3% to 85.5%, Ø 11.8%; PE_ODV_ −0.7% to 19.5%, Ø −2.68%) also show a good predictive power of the model.

The PBPK model detected a high variability in VEN’s pharmacokinetics, which is due to CYP polymorphisms. Simulation of extensive and poor metabolizers showed acceptable precision and accuracy. Because there was no investigation of CYP polymorphisms in KONBEST, a k_cat_ for CYP2D6 describing plasma concentrations in populations without knowledge about their pheno- or genotypes was used for DDI models (k_cat_ = 29.16 min^−1^) due to its high accuracy under steady-state conditions.

Also, BUP’s pharmacokinetics were well described by the created model with single and multiple drug administrations. The pharmaceutical formulation, which is also commercially available with extended release, was accurately simulated with a mean t_max_ of 5.2 ± 0.6 h (range of PE = −8.95 to 11.1%), which is in accordance with the prescribing information [10].

The extensive metabolization of BUP to OHB, THB and EHB was well described by the PBPK model. The percentage of unchanged drugs in the urine is in accordance with the described amount in the literature [27]. Pharmacokinetics were predicted precisely with an acceptable mean bias (MPE BUP, THB < 30%, EHB < 45%, under steady-state conditions < 20%) and a good mean precision (MAPE < 50%, under steady-state conditions < 25%), especially under steady-state conditions. Accurate PE in C_max_ (PE_BUP_ −45.4% to 3.21%, Ø −23.5%; PE_OHB_ −12.5% to 46.0%, Ø 3.20%; PE_THB_ −24.1% to 12.6%, Ø −7.83%; PE_EHB_ −14.9% to 50.5%, Ø 13.4%) and AUC (PE_BUP_ −34.7% to 29.8%, Ø −14.8%; PE_OHB_ −19.3% to 60.6%, Ø 4.27%; PE_THB_ −27.8% to 108%, Ø 7.12%; PE_EHB_ −7.81% to 70.0%, Ø 19.6%) underline the good predictive power of the model.

The GOF plot (Figure 3) illustrates that 60.0% (steady state of 77.5%) of all plasma concentrations are in 1.25-fold and 95.1% (steady state of 100%) are in the 2-fold error range. The PBPK model also detected a high variability in BUP’s pharmacokinetics, which is mainly due to CYP and UGT polymorphisms. No drug–gene interaction simulation was performed, because there were no data found describing the pharmacokinetics of the extended-release formulation of BUP in pheno- or genotyped patients.

### 3.2. DDI Model in Younger and Older Patients

The inhibitory potential on CYP2D6 by BUP, OHB, THB and EHB on the metabolism of VEN to ODV is accurately represented by the DDI model. This is demonstrated by the proportions of the C_tr_ of VEN, ODV and AM falling within the 5th to 95th percentile of the model calculated by PK-Sim^®^ (Table 4).

Figure 4 illustrates that as soon as BUP comedication is started, the concentration of VEN increases. At the same time, the concentration of ODV is reduced. Meanwhile, the AM remains the same throughout the course of the treatment. This can be observed in both younger and older patients and occurs both with the administration of 150 mg and 300 mg BUP.

Especially in young patients treated concomitantly with 150 mg BUP, the PBPK model displays a good predictive power. The model shows a slight underprediction of C_tr_ < 20% in this group (Table 4), whereas the MR_ODV/VEN_ was predicted accurately with a low PE of 4.94%. In contrast, young patients treated with 300 mg BUP show a higher variability in plasma C_tr_ of VEN and ODV. This is illustrated in a lower percentage of KONBEST data falling within the 5th to 95th percentile. Also, the underprediction of C_tr_ VEN and AM is more pronounced than in young patients treated with 150 mg BUP. Contrarily, the C_tr_ of ODV and MR_ODV/VEN_ are overpredicted.

In elderly patients, a similar situation is emerging: The PE except for MR_ODV/VEN_ is <40%. Unfortunately, only a little data are available from KONBEST, but, nevertheless, a high variability in C_tr_ is detectable (Table 4). A comparison of the median C_tr_ of older and younger patients shows that these are increased in older patients regardless of the BUP dose. If the median KONBEST C_tr_ of younger and older patients are compared with each other, percentage differences can be determined, which appear to vary depending on the BUP dose. For example, the C_tr_ of VEN in older patients after administration of 150 mg BUP is approximately 80% higher than in younger patients, and, after administration of 300 mg, the difference is only 10% higher. This may be due to the wide variation in the data. In contrast, the PBPK model remains constant at approximately 30% higher C_tr_ in elderly patients regardless of the BUP dose. Surprisingly, the MR_ODV/VEN_ is 25% lower in older patients after administration of 150 mg BUP, while it remains constant after administration of 300 mg BUP (Table 5).

Compared to younger patients, the AUC of VEN, ODV and AM is higher in elderly patients. However, the rate of the AUC rise under comedication with BUP remains stable regardless of the dose in older patients: The AUC of AM in older patients after administration of 150 and 300 mg BUP is approximately 17% higher than in younger patients and, without simultaneous administration of BUP, it is 15% higher. The MR_AUC_ is negligibly reduced in the elderly. The simultaneous administration of bupropion also only leads to an irrelevant reduction in MR_AUC_ of 14%, without any change when the dose of BUP is increased (Table 4). A dose reduction in BUP because of age seems to be not necessary. The dosage should therefore be following the prescribing information. To emphasize the age-related changes, the Appendix A show that AUC and C_max_ are slightly higher in older patients and support the statements made previously.

Considering the illustration of the plasma concentration–time curves (Figure 4), it is apparent that, in general, concentrations of VEN, ODV and AM are marginally higher in elderly patients. In particular, the comparison of the change in the AUC of VEN, ODV, AM and MR_AUC_ before and during concomitant use of BUP indicates no major age-related difference. Even if there is about 25% lower MR_AUC_ in patients treated with 300 mg BUP compared to the intake of 150 mg BUP, no clinically relevant change in AM is detectable regarding age and dose of the perpetrator drug BUP (Table 5). This is also underlined by the fact that during comedication with BUP, the C_tr_ of AM remains in the therapeutic reference range (100–400 ng/mL) regardless of the patient’s age and BUP dose (Figure 4). These results indicate that no dose adjustment is required for concomitant administration of venlafaxine and bupropion. MDDIcalc computes that there is no impact on the AUC of AM in the context of DDI with BUP and does not recommend any dose adjustment.

### 3.3. Multiple Drug–Drug Interaction of VEN, BUP and ITRA

Finally, the dual interaction model was supplemented with the ITRA template from PK-Sim^®^ to simulate a theoretical consideration in which it is investigated how the plasma concentrations and AUC of VEN, ODV and AM change if, in addition to CYP2D6 inhibition by BUP, there is additional CYP3A4 inhibition by ITRA.

The plasma concentration–time curves indicate that during MDDI with ITRA and BUP, the AM arises (Figure 5). Especially in elderly patients, AM exceeds the therapeutic reference range, which is associated with a higher risk of ADR. Therefore, in individual cases, there may be a need to reduce the dose of VEN or to change the antifungal agent. The plasma concentration–time curves of the DDI VEN + ITRA are shown in the Appendix A (Appendix A).

The exposure (AUC) of VEN and ODV is increased by approximately 12% compared to DDI with BUP. This is the case for both younger and older patients. The AUC of AM, which is used to assess the clinical relevance of the interaction, also shows an increase of 12% compared to the DDI. Compared with the simultaneous use of VEN and ITRA without BUP comedication, the AUC of the AM only increases by 3% and 5.73% in older patients. In individual cases, it may therefore be necessary to adjust the therapy as previously stated (Table 6).

No recommendation can be made, as this is merely a “what-if” simulation without plasma concentrations available to verify the accuracy and predictive power of the model. The MDDIcalc indicated that the AUC of AM increased by 51% during the concomitant use of BUP and ITRA. For the DDI VEN/ITRA, the MDDIcalc computes a change in the AUC of VEN of about +24%.

## 4. Discussion

The implementation of the inhibitory influence of BUP and its metabolites OHB, THB and EHB was successful. Currently, there is no PBPK model for the interaction of VEN and BUP that integrates ODV into the calculations of the inhibition’s extent. Furthermore, the DDI model developed is the first of its kind. The DDI model created by Xue et al. [42] with Simcyp^TM^ does not consider the behavior of ODV during the interaction and the inhibition by metabolically formed OHB, THB and EHB. Also, they administered the metabolites of BUP orally to the model instead of producing them from the parent substance via metabolism [42]. In PK-Sim^®^, there is also currently no DDI approach for VEN and BUP.

The K_i_ values of BUP, OHB, THB and EHB calculated using PK-Sim^®^ are consistent with the C_tr_ calculated using a combination of VEN and BUP. During parameter identification, it was considered that the inhibitory potential of BUP, OHB, THB and EHB determined in vitro by Reese et al. [14] has the same rank order.

The underestimation of C_tr_ (VEN) and C_tr_ (AM) after simulated administration of 300 mg BUP is greater than in the case of 150 mg BUP. This could be due to a greater scattering of the extracted data, as can also be seen in Figure 4, or to the low data density. However, the majority of C_tr_ extracted from KONBEST are in the 5th to 95th percentile, confirming the good predictive power of the developed DDI model.

According to EMA [4] and FDA [16] guidelines, the extent of an interaction is defined as strong if the AUC of the affected substrate is increased more than fivefold by simultaneous administration of an inhibitor. In the simulated interaction of VEN and BUP, the AUC of VEN is increased by a factor of 2.1 to 2.3. This would correspond to a moderate inhibition. The more sensitive response of (R)-VEN to strong CYP2D6 inhibition compared to (S)-VEN [55] could be a reason for the deviation from the EMA or FDA data. When racemic VEN is used, the differing inhibition of its individual enantiomers may result in a weaker overall effect. In contrast, non-racemic substances like dextromethorphan and desipramine, which are FDA and EMA clinical index/probe drugs for CYP2D6 inhibition studies [4,16], provide a clearer basis for assessing CYP2D6 inhibition. This difference may explain why the FDA and EMA criteria for evaluating strong CYP2D6 inhibitors like BUP are not fully applicable in the case of VEN. Thus, the inhibitory potential of BUP in the racemate could be reduced compared to the VEN enantiomers considered individually. The involvement of other CYP enzymes (CYP2C9, CYP2C19 and CYP3A4) in the metabolism of VEN to ODV [7] could also compensate for the inhibitory effect of BUP, meaning that the criterion of strong inhibitory potential is not met.

The C_tr_ (VEN) extracted from KONBEST was not determined stereoselectively due to the exclusively racemic use. An investigation of the effect of BUP on the enantiomers of VEN could be a future approach to further investigate the nature of the interaction. As(R)-VEN acts as a noradrenaline and serotonin reuptake inhibitor, (S)-VEN only influences serotonin reuptake, the enantiomers differ in their effect [56]. The interaction could therefore be expected to enhance noradrenergic ADRs like higher blood pressure or cardiovascular ADRs, so that the stereoselective consideration could also represent a relevant future factor in addition to a pharmacodynamic consideration of the DDI. For example, PK/PD modeling could be used for this purpose in order to relate blood pressure to the interaction with BUP in addition to the recorded plasma levels of stereoselective VEN and ODV. Similarly, multiple and accurate sampling times and simultaneous concentration determination of BUP, OHB, THB and EHB, as well as VEN and ODV, could confirm the good predictivity of the model concerning inhibitory potential.

The simultaneous increase in the exposure of VEN and reduction in ODV with approximately the same AM under BUP comedication, presumably, does not affect the therapeutic effect. This is also demonstrated by the DDI model: The AUC changes calculated there show that despite the increase in the AUC of VEN due to the decrease in the AUC of ODV, ultimately, the AUC of AM remains the same or decreases negligibly (<3%). This is consistent with the statements in the prescribing information [6] that similar overall exposure is observed with poor (PM) and extensive metabolizers (EM) and, therefore, no dose adjustment is necessary. A strong CYP inhibitor phenoconverts an EM to a PM [57] and its influence can therefore be equated with this observation.

The extrapolation to a collective of older patients confirms the adequate transferability of the developed PBPK DDI model for patients ≥ 65 years. Although the simulations estimate C_tr_ (VEN) and C_tr_ (ODV) to be larger in relation to younger patients, the differences in the medians are small. However, the small number of patients available for model evaluation should be noted (comedication with 150 mg BUP *n* = 8, 300 mg *n* = 5). In addition to the variability of the C_tr_ (VEN) and C_tr_ (ODV), this small number of patients could explain the deviation of the MR_ODV/VEN_ between the median of KONBEST data and the estimation by the PBPK model at a comedication of 300 mg BUP. Furthermore, it cannot be ruled out that the patients had a history of kidney or liver dysfunction. In these patients in particular, the prescribing information [10] recommends reducing the dose to 150 mg BUP due to the risk of accumulating metabolites. The cumulative metabolites OHB, THB and EHB could increase the inhibitory potential, resulting in higher plasma concentrations of VEN. This might not be detectable by the model, as it has not been validated for patients with renal and/or hepatic impairment. In the future, values such as creatinine levels and GFR or the status of liver failure could also be taken into account when collecting data in the context of Therapeutic Drug Monitoring (TDM). This would allow the model to be extrapolated to a patient group with, e.g., renal insufficiency, and the extent of the interaction to be re-evaluated.

Even if the interaction appears clinically irrelevant according to the observations of the PBPK simulation, it should not be underestimated. The use of CYP2C19-PM, for example, could lead to toxic effects according to Kringen et al. [58]. TDM is therefore still recommended for the simultaneous application to ensure patient safety and to be able to quickly counteract an increase in AM, which is associated with an increased risk of ADR. The prediction that there is no change in the AUC of AM by the MDDIcalc may suggest to users of the SDB that no dose adjustment is recommended when VEN and BUP are administered concomitantly, from a pharmacokinetic perspective. However, it should not be taken as an indication that there is no pharmacokinetic interaction at all.

The effect of a strong CYP3A4 inhibitor on the previously simulated interaction of VEN and BUP was additionally modeled with ITRA as a template of PK-Sim^®^. It is not possible to verify the results obtained due to a lack of data, but conclusions can be drawn from the simulations on the effect of dual inhibition on the metabolism of VEN. Compared to the pair interaction of VEN and BUP, the MDDI with ITRA calculated an additional increase in AUC in younger patients of VEN, ODV and AM by 12.3%, 11.9% and 12.3% (older patients: +12.4%, +12.0%, +12.2%). Despite an age-independent increase in the AUC of VEN by 2.7-fold, corresponding to moderate inhibition, there was only a slight increase in the AUC of AM, which is not classified as clinically relevant. However, toxic reactions may still occur in individual patients, particularly in the presence of renal and liver dysfunction. The prescribing information of Effexor XR [6] advises caution in the concomitant use of VEN and strong CYP3A4 inhibitors.

The AUC increase in AM calculated by MDDIcalc (51%) differs from the calculation of the MDDI-PBPK model. This could be related to the ITRA template used from PK-Sim^®^. For example, the template only considers the metabolite hydroxy-ITRA in addition to ITRA, but its metabolites keto- and N-desalkyl-ITRA also exert an inhibitory influence on CYP3A4 [59]. There are also indications that the inhibitory effect of ITRA may be greater in reality than predicted in the PBPK model. Park et al. [60] found that administration of ITRA in CYP2D6 PM increased the AUC of CYP2D6 and 3A4 substrate haloperidol three-fold. Since a strong CYP2D6 inhibitor like BUP achieves a phenoconversion from EM to PM, this result can be transferred to VEN (a 2.7-fold increase in AUC (VEN) in the MDDI model).

Lindh et al. [61] demonstrated that when the strong CYP3A4 inhibitor ketoconazole (KETO), which also moderately inhibits CYP2C19, was used with CYP2D6-PM, AUC of VEN and ODV increased by 70% and 21.3%, respectively. This is consistent with the calculations of the MDDIcalc, which calculated an increase in the AUC of AM by about 54% (compared to the 51% in the ITRA case). It should be noted that ITRA has a lower inhibitory potential on CYP3A4 than KETO. Wang et al. [62] were able to demonstrate a 13-fold lower K_i_ value (KETO = 0.18, ITRA = 2.30 μM) for inhibition of CYP3A4-associated midazolam metabolism and thus an increased inhibitory potential. Yamaguchi et al. [63] also found that ITRA has at least a ten-fold weaker inhibitory effect on testosterone metabolism via CYP3A4 than KETO.

The findings from these in vitro studies and the calculations by MDDIcalc show that the different results can only be confirmed and assessed with the help of the missing plasma concentrations of VEN and ODV, as well as the AM in the MDDI setting.

Overall, MDDIcalc is a helpful tool that can be used to predict the effect of several inhibitors of CYP enzymes on the exposure of a substrate. It should be noted that the MDDIcalc is limited in the presence of non-linear pharmacokinetics, a restriction of minor importance as in the therapeutic dose range, the metabolism of drugs is normally subject to first-order kinetics [64]. It is furthermore limited when an age-dependent change in pharmacokinetics occurs and in the presence of active metabolites when only aggregating results. Integration of computing in the case of prodrugs, specifically, the exposures of ingredient and active metabolites into the unpublished calculation formula of MDDIcalc, would be a conceivable future development and would improve the validity of the calculator. It must be considered that the calculations made always represent an orientation value that must be critically scrutinized by the user. The statements made by the MDDIcalc about a dose reduction must always be individually adapted to the patient [54].

An MDDI-PBPK model enables the integration of active metabolites and non-linear pharmacokinetic processes. The patient to be simulated can also be individualized concerning demographics (age, weight, height, biological sex, BMI) and the given dose of a drug. Thus, these limitations of SDB’s MDDIcalc can be circumvented by the PBPK model. However, PBPK simulations always require a high level of a priori knowledge of substances, metabolites and patients. Adjustments to the model concerning factors like renal and liver dysfunction and the presence of genetic polymorphisms must frequently be evaluated with concentration data from patients, which are not always available. Here, SDB’s MDDIcalc offers a simple approach to efficiently predict AUC changes. SDB’s MDDIcalc does not require the search of all parameters required for PBPK modeling and provides an overview of how the exposure of substrates changes in a patient with polymedication if inhibitor drugs are used therapeutically.

Finally, this comparison shows that both a calculation of the change in exposure in the MDDI process using SDBs MDDIcalc and PBPK is a way to continuously improve individual pharmacotherapy. Nevertheless, continuous further development is urgently recommended so that the predictive power is continuously optimized.

## 5. Conclusions

Besides polymorphisms of CYPs (especially 2D6, 2C19 and 2B6) and UGTs, factors like renal and hepatic insufficiency are relevant for the individual clinical assessment and should be considered in the further development of the PBPK model, as these were not considered in the KONBEST data sampling.

The ITRA model of PK-Sim^®^ was not modified, so the results from the MDDI model are merely a guide and are suitable for clinical application. MDDIcalc demonstrated good consistency with in vivo drug exposure data in the multiple interaction scenario of VEN–BUP–ITRA. In this respect, future generation of data, e.g., in the context of TDM, is required to verify the validity of the MDDI-PBPK model. These data could be used to assess the clinical relevance of the interaction.

## Figures and Tables

**Figure 1 pharmaceutics-17-00179-f001:**
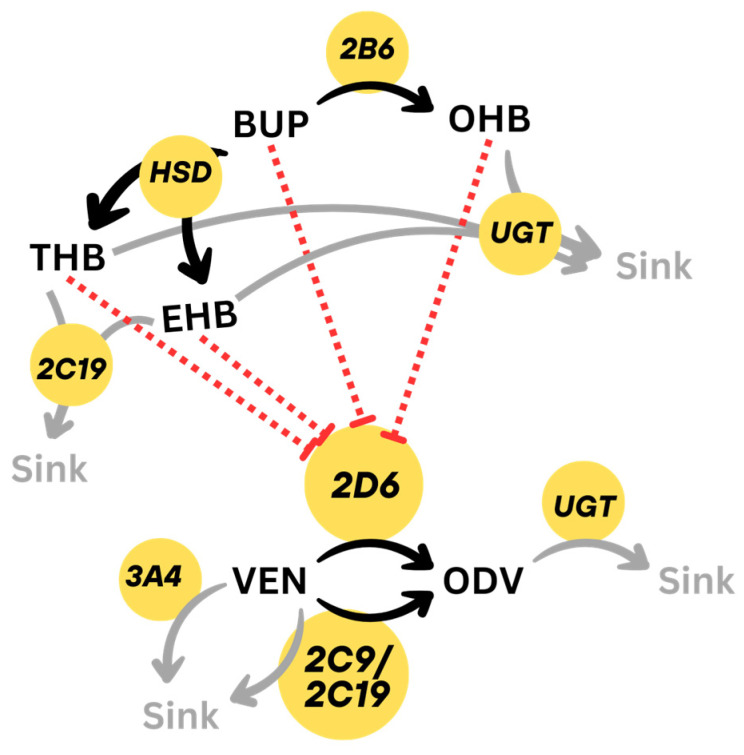
Schematic overview of the metabolism integrated into PBPK model. BUP: bupropion, EHB: erythrohydrobupropion, ODV: O-desmethylvenlafaxine, OHB: hydroxybupropion, Sink: not defined metabolite(s), THB: threohydrobupropion, UGT: uridinglucoronosyltransferase, VEN: venlafaxine. Red dashed lines show inhibitory effects, black arrows illustrate main metabolic pathways, while gray arrows demonstrate minor metabolic pathways.

**Figure 2 pharmaceutics-17-00179-f002:**
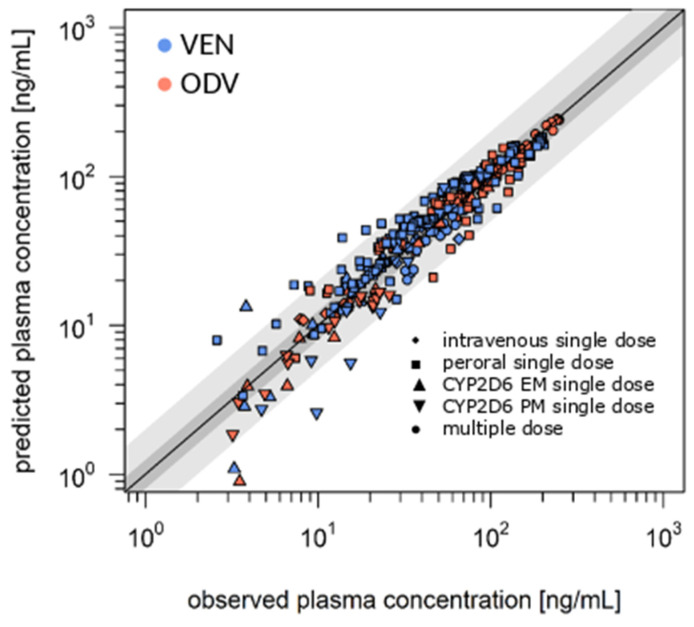
GOF plot of VEN literature model. The dark gray area in the GOF plot illustrates the 1.25-fold error range and light gray areas show the 2-fold error range. EM: extensive metabolizer, ODV: O-desmethylvenlafaxine, PM: poor metabolizer, VEN: venlafaxine.

**Figure 3 pharmaceutics-17-00179-f003:**
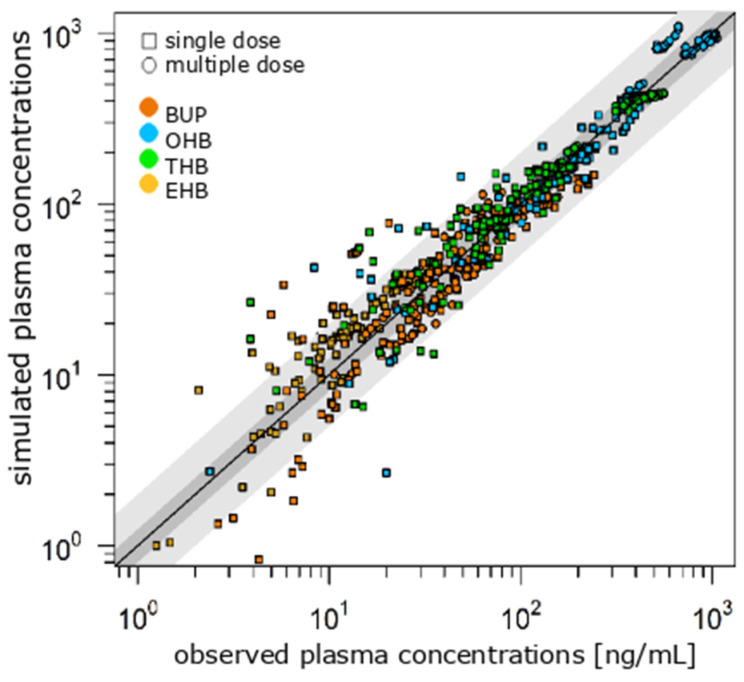
GOF plot of literature-based BUP PBPK model. The dark gray area in the GOF plot illustrates the 1.25-fold, the light gray area the two-fold error range. BUP: bupropion, EHB: erythrohydro-bupropion, OHB: hydroxybupropion, THB: threohydrobupropion.

**Figure 4 pharmaceutics-17-00179-f004:**
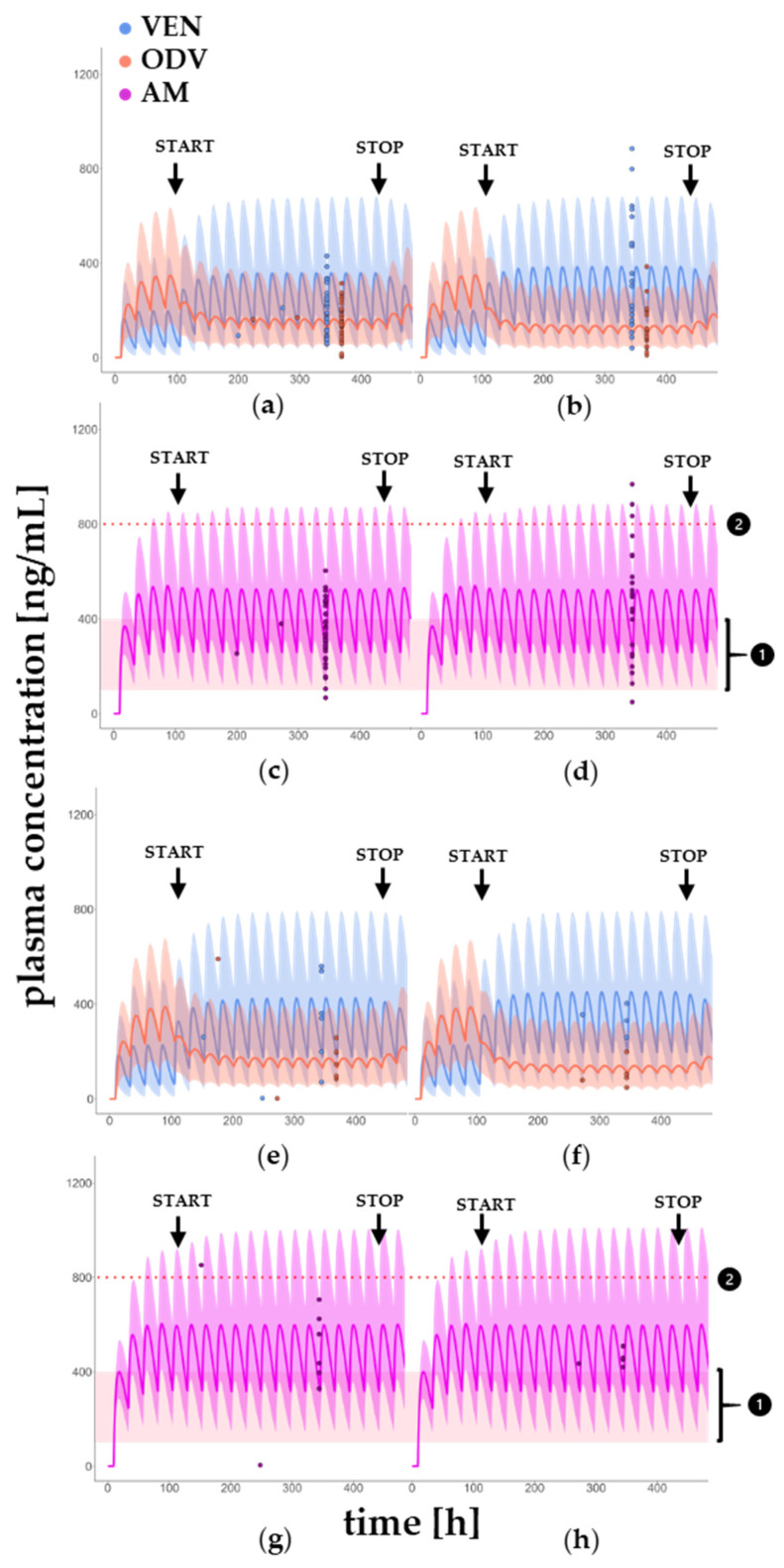
Simulation of DDI in young and elderly patients. Plasma concentration–time curves of VEN, ODV (**a**,**b**,**e**,**f**) and AM (**c**,**d**,**g**,**h**) with concomitant use of 150 mg (**a**,**c**,**e**,**g**) and 300 mg BUP (**b**,**d**,**f**,**h**) in young (**a**–**d**) and elderly (**e**–**h**) patients. Measured ODV plasma concentrations were offset by 24 h for the overview in the figure. However, the determination was made from the same sample as that of VEN. The plasma concentration–time courses of BUP, OHB, THB and EHB were not shown for reasons of clarity.1 in black circle: Therapeutic reference range, 2 in black circle: toxic range, START: start of BUP, STOP: end of BUP use.

**Figure 5 pharmaceutics-17-00179-f005:**
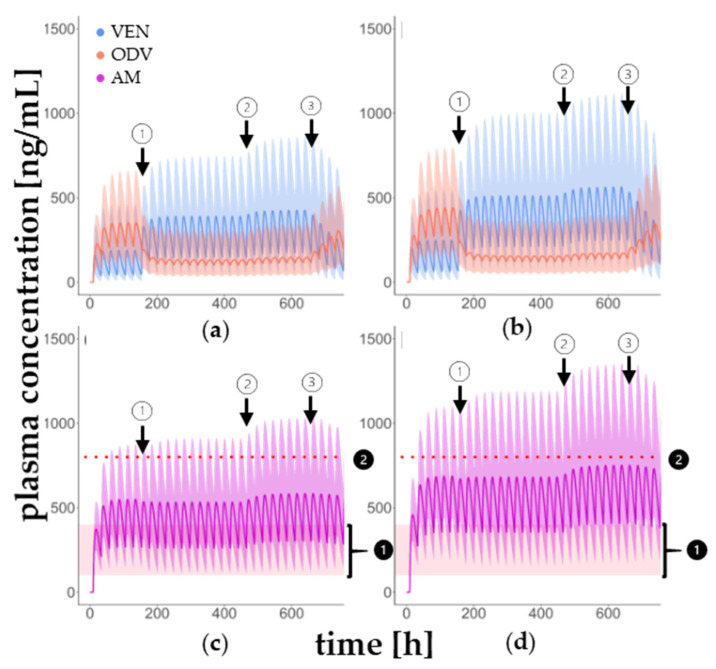
Plasma concentration–time curves of VEN, ODV and AM during MDDI in young (**a**,**c**) and elderly (**b**,**d**) patients. Administration protocol contains 1 in white circle: start of 300 mg BUP once daily, 2 in white circle: start of 100 mg ITRA twice daily and 3 in white circle: stop BUP and ITRA administration. To interpret the influence of MDDI, parameters of TDM (1 in black circle: therapeutic reference range (AM 100–400 ng/mL), 2 in black circle: toxic range (AM > 800 ng/mL)) are used.

**Table 1 pharmaceutics-17-00179-t001:** Inhibitoric constants K_i_ used to characterize CYP2D6 inhibition caused by BUP and metabolites.

Substance	K_i_ (Model)	K_i_ (Literature)	Reference
bupropion	0.46 µM	21.0 µM	[14]
hydroxybupropion	0.41 µM	13.3 µM	[14]
threohydrobupropion	0.15 µM	5.40 µM	[14]
erythrohydrobupropion	0.04 µM	1.70 µM	[14]

**Table 3 pharmaceutics-17-00179-t003:** Patient characteristics extracted from KONBEST.

Dose VEN [mg]	Dose BUP [mg]	N	% Female (♂/♀)	Age [Years]	Height [cm]	Weight [kg]	BMI [kg/m^2^]
Young patients							
225 ^†^	150 300	35 22	58.0 (24/33)	50 (20–63)	171 (152–193)	90 (52–140)	29 (19–48)
Elderly patients							
255 ^†^	150 300	8 5	53.8 (6/7)	69 (65–78)	166 (160–181)	78 (65–112)	28 (24–39)

^†^ dose normalized to 225 mg VEN, BMI: body mass index, BUP: bupropion, N: number of subjects, VEN: venlafaxine.

**Table 4 pharmaceutics-17-00179-t004:** Comparison of KONBEST data and simulated plasma concentrations in young and elderly patients.

Dose_BUP_ [mg]	Parameter	N	%_5–95_ Percentile	Data Median (min–max)	Model Median (min–max)	PE_median_ [%]
young patients						
150	C_VEN_	35	94.3	169 (56.3–430)	144 (15.8–731)	−14.8
	C_ODV_	35	88.6	149 (2.25–315)	126 (20.3–527)	−15.4
	C_AM_	35	91.4	335 (67.5–603)	277 (54.0–1051)	−17.3
	MR_ODV/VEN_	35	-	0.81 (0.02–2.87)	0.85 (0.72–1.29)	4.94
300	C_VEN_	21	52.4	322 (38.3–882)	162 (18.0–1013)	−49.7
	C_ODV_	21	76.2	87.8 (11.3–405)	108 (20.3–441)	23.0
	C_AM_	21	66.7	500 (49.5–968)	281 (81.0–1177)	−43.8
	MR_ODV/VEN_	21	-	0.26 (0.02–2.78)	0.63 (0.44–1.13)	142
elderly patients						
150	C_VEN_	8	62.5	302 (2.25–560)	189 (18.0–878)	−37.4
	C_ODV_	8	75.0	171 (2.25–592)	140 (18.0–599)	−18.1
	C_AM_	8	75.0	497 (4.50–853)	340 (40.5–1031)	−31.6
	MR_ODV/VEN_	8	-	0.61 (0.16–3.63)	0.74 (0.68–1.00)	21.3
300	C_VEN_	5	100	356 (261–403)	214 (22.5–891)	−39.9
	C_ODV_	5	100	90.0 (47.3–198)	117 (13.5–518)	30.0
	C_AM_	5	100	452 (421–509)	340 (36.0–1053)	−24.8
	MR_ODV/VEN_	5	-	0.26 (0.12–3.77)	0.55 (0.58–0.60)	112

-: Not examined, AM: active moiety, BUP: bupropion, C: concentration, MR: metabolic ratio, N: number of samples, ODV: O-desmethylvenlafaxine, PE: prediction error, VEN: venlafaxine.

**Table 5 pharmaceutics-17-00179-t005:** Change in AUC during concomitant use of 150 mg and 300 mg BUP and comparison of age-relevant change in the AUC.

AUC_VEN_ [ng·h/mL]			AUC_+150 mg BUP_ [ng·h/mL]		AUC_+300 mg BUP_ [ng·h/mL]		AUC_300_/AUC_150_	
young	VEN ODV AM	2949 7086 10,240	6197 3543 10,031	+110% −50.0% −2.04%	6809 2935 9994	+131% −58.6% −2.40%	1.10 0.83 1.00	+9.88% −17.2% −0.37%
	MR_AUC_	2.40	0.57	−76.3%	0.43	−82.1%	0.75	−24.6%
old	VEN ODV AM	3578 8047 11,809	7681 3775 11,689	+115% −53.1% −1.02%	8297 3097 11,679	+132% −61.5% −1.10%	1.08 0.82 1.00	+8.02% −18.0% −0.09%
	MR_AUC_	2.25	0.49	−78.2%	0.37	−83.6%	0.76	−24.5%
AUC_old_/AUC_young_								
	VEN	1.21	+21.3%	1.24	+23.9%	1.22	+21.9%	0.98
	ODV	1.14	+13.6%	1.07	+6.55%	1.06	+5.52%	0.99
	AM	1.15	+15.3%	1.17	+16.5%	1.17	+16.9%	1.00
	MR_AUC_	0.94	−6.25%	0.86	−14.0%	0.86	−14.0%	1.01

AM: active moiety, AUC: area under the curve, BUP: bupropion, MR: metabolic ratio, ODV: O-desmethylvenlafaxine, VEN: venlafaxine.

**Table 6 pharmaceutics-17-00179-t006:** AUC, C_max_ and c_min_ of the MDDI simulation’s medians of VEN, ODV and AM.

AUC_VEN_ [ng·h/mL]			AUC_VEN+BUP_ [ng·h/mL]		AUC_VEN+ITRA_ [ng·h/mL]		AUC_VEN+BUP+ITRA_ [ng·h/mL]		AUC_MDDI_/ AUC_VEN+BUP_		AUC_MDDI_/ AUC_VEN+ITRA_	
young	VEN ODV AM	2822 7131 10,441	6838 2913 10,128	+142% −59.1% −3.00%	2969 7507 11,044	+5.23% +5.27% +5.78%	7681 3258 11,371	+172% −54.3% +8.91%	1.12 1.12 1.12	+12.3% +11.8% +12.3%	2.59 0.43 1.03	+159% −56.6% +3.00%
old	VEN ODV AM	3845 9104 13,269	9199 3441 13,205	+139% −62.2% −0.48%	4048 9485 14,017	+5.26% +5.23% +5.64%	10,339 3854 14,820	+169% −57.7% +11.7%	1.12 1.12 1.12	+12.4% +12.0% +12.2%	2.55 0.41 1.06	+155% −59.4% +5.73%
Cmax_VEN_ [ng/mL]			Cmax_VEN+BUP_ [ng/mL]		Cmax_VEN+ITRA_ [ng/mL]		Cmax_VEN+BUP+ITRA_ [ng/mL]		Cmax_MDDI_/ Cmax_VEN+BUP_		Cmax_MDDI_/ Cmax_VEN+ITRA_	
young	VEN ODV AM	189 352 549	390 132 533	+106% −62.5% −2.91%	197 367 578	+4.32% +4.26% +5.28%	426 147 584	+125% −58.2% +6.38%	1.09 1.11 1.10	+9.23% +11.4% +9.57%	2.16 0.40 1.01	+116% −59.9% +1.04%
old	VEN ODV AM	248 440 687	512 155 682	+106% −64.8% −0.73%	260 458 719	+4.84% +4.09% +4.66%	565 173 751	+128% −60.7% +9.32%	1.10 1.12 1.10	+10.4% +11.6% +10.1%	2.17 0.38 1.04	+117% −48.1% +4.45%
Cmin_VEN_ [ng/mL]			Cmin_VEN+BUP_ [ng/mL]		Cmin_VEN+ITRA_ [ng/mL]		Cmin_VEN+BUP+ITRA_ [ng/mL]		Cmin_MDDI_/ Cmin_VEN+BUP_		Cmin_MDDI_/ Cmin_VEN+ITRA_	
young	VEN ODV AM	48.3 227 288	161 107 281	+233% −52.9% −2.43%	51.7 237 307	+7.04% +4.41% +6.60%	190 123 327	+293% −45.8% +13.5%	1.18 1.15 1.16	+18.0% +15.0% +16.4%	3.68 0.52 1.07	+268% −48.1% +6.51%
old	VEN ODV AM	70 293 382	231 127 380	+230% −56.7% −0.52%	75 307 406	+7.14% +4.78% +6.28%	268 146 435	+283% −50.2% +13.9%	1.16 1.15 1.14	+16.0% +15.0% +14.5%	3.57 0.48 1.07	+257% −52.4% +7.14%

Deviations in % (gray columns) from the AUC, C_max_ and C_min_ in the steady state of VEN administration alone. Deviations in % (white columns) show the difference between the settings plotted in the quotients. AM: active moiety; AUC: area under the curve; BUP: bupropion; Cmax: maximal concentration in steady state; Cmin: minimal concentration in steady state; ITRA: itraconazole; MDDI: multi-drug–drug interaction (VEN + BUP + ITRA); ODV: O-desmethylvenlafaxine; VEN: venlafaxine.

## Data Availability

The data and model files are available on reasonable request from the corresponding author. The request should be accompanied by a research protocol. The data are not publicly available due to European ethical and legal restrictions.

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
