# Peer review of "Physiologically Based Pharmacokinetic Model of CYP2D6 Associated Interaction Between Venlafaxine and Strong Inhibitor Bupropion—The Influence of Age-Relevant Changes and Inhibitory Dose to Classify Therapeutical Success and Harm"

_pharmaceutics, 2025, doi:10.3390/pharmaceutics17020179_

Round 1
Reviewer 1 Report
Comments and Suggestions for Authors
The manuscript is well written and there are no issues concerning the methodology and data presentation.
Throughout the manuscript there are similar multiple typos such as cmax or ctr where all letters are written in subscript.
Author Response
The manuscript is well written and there are no issues concerning the methodology and data presentation. Throughout the manuscript there are similar multiple typos such as cmax or ctr where all letters are written in subscript.
Thank you for your valuable feedback. We appreciate your careful review of the manuscript. Regarding the typographical issues you pointed out with terms like "Cmax" and "Ctr," we have addressed this in the revised manuscript. We ensured that the formatting is consistent and accurate throughout, with only the appropriate letters in subscript. Your attention to detail has been very helpful, and we believe the corrections have improved the overall clarity of the manuscript.

Reviewer 2 Report
Comments and Suggestions for Authors
The authors developed a PBPK model to predict drug interaction between venlafaxine and bupropion. The methodology was clear and rigorous. I have only a few comments, as outlined below.
1. Please specify which model i.e., flow-limited, or permeability-limited was used for the PBPK model.
2. Provide a more detailed explanation of why the competitive inhibition model was selected for this study.
3. Do you consider performing a sensitivity analysis to identify the most influential parameters in the model?
Author Response
The authors developed a PBPK model to predict drug interaction between venlafaxine and bupropion. The methodology was clear and rigorous. I have only a few comments, as outlined below.
Thank you very much for your thoughtful review of our manuscript. We greatly appreciate your kind words regarding the clarity and rigor of the methodology. We are grateful for the time and effort you have dedicated to reviewing our work.
- Please specify which model i.e., flow-limited, or permeability-limited was used for the PBPK model.
Thank you for your helpful suggestion. The PBPK model used in our study is flow-limited, and we have now explicitly mentioned this in the revised manuscript for clarity
- Provide a more detailed explanation of why the competitive inhibition model was selected for this study.
Thank you for your insightful comment. The selection of the competitive inhibition model was based on the findings from the study by Reese et al. [14], which demonstrated in vitro that the inhibition of CYP2D6 by bupropion and its active metabolites is likely competitive. To our knowledge, no further investigations into this mechanism have been conducted since then. We have now referenced this study in the manuscript to provide additional context for our choice of the inhibition model.
- Do you consider performing a sensitivity analysis to identify the most influential parameters in the model?
Thank you for your suggestion regarding a sensitivity analysis. While we fully recognize the value of such an analysis for identifying influential parameters in the model, time constraints and project
priorities played a significant role in this case. Our primary focus was on calibrating and validating the PBPK model to ensure it aligns closely with experimental and literature data. Additionally, considerable effort was directed toward managing and organizing the necessary input data, which required substantial resources. Given these circumstances, we decided to prioritize these steps over
conducting a sensitivity analysis. However, we appreciate your feedback and will certainly consider incorporating this in future work or follow-up analyses when resources allow.

Reviewer 3 Report
Comments and Suggestions for Authors
This manuscript presents a physiologically based pharmacokinetic (PBPK) model to study the interaction between venlafaxine (VEN) and bupropion (BUP) and its impact on pharmacokinetics in different age groups. The study is relevant as drug-drug interactions are common in pharmacotherapy, especially in treating depression with polypharmacy. The manuscript is generally well-written, with clear objectives, methods, results, and discussion sections. The use of PBPK modeling and comparison with the MDDI calculator is a strength. However, some areas could be improved for better clarity and comprehensiveness.
Abstract
Could provide more specific details about the age-relevant changes in the abstract, such as the differences in AUC changes between younger and older patients.
The mention of the need for further research regarding elderly patients, renal or hepatic impairment, and polymorphisms is very brief and could be elaborated on.
Introduction
While it mentions the importance of individual pharmacokinetics, it could further discuss how the current study will contribute to personalized medicine in this context.
The introduction could briefly touch on the current state of knowledge regarding the VEN-BUP interaction and what gaps the current study aims to fill.
Materials and Methods
Some of the technical details, such as the specific equations used in PK-Sim for estimating drug-associated models, could be more clearly presented or referenced for readers who want to understand the modeling process in depth.
The description of the multiple interaction simulation with itraconazole could be more detailed, especially regarding the rationale for choosing this combination and the expected outcomes.
Results
In the DDI model results, while the differences in ctr and AUC between younger and older patients are described, a more in-depth statistical analysis could be performed to determine the significance of these differences.
The interpretation of the results regarding the clinical relevance of the DDIs could be more explicit, especially in relation to the therapeutic reference range and the potential need for dose adjustment.
Discussion
Some of the explanations, such as the possible reasons for the deviation from the EMA or FDA data on inhibitory potential, could be further supported by additional references or in vitro studies.
The discussion on future research directions could be more specific, for example, outlining the exact types of studies needed to address the limitations of the current model and improve the understanding of the VEN-BUP interaction.
The discussion on the potential effect of BUP on the enantiomers of VEN could be expanded. Future research directions regarding the stereoselective consideration of the interaction and its pharmacodynamic implications could be more clearly outlined.
Minor issues:
Page 11, Table 4: In the “parameter” column, “CVEN” and “CODV” might be more intuitive if written as “CVEN” and “CODV” to clearly indicate they represent the concentrations of VEN and ODV.
Throughout the manuscript: Some of the abbreviations are introduced and then used without always being fully spelled out again in subsequent sections where they might not be as immediately familiar to the reader. For example, after first introducing “TDM” as “therapeutic drug monitoring,” it could be beneficial to briefly remind the reader of the full term when it is used again in a different context later in the paper.
Figure captions: While the figures are generally helpful, some of the captions could provide more detailed explanations. For instance, in Figure 4, in addition to stating the doses of BUP and the age groups, it could briefly mention what the overall trends or key takeaways from the plasma concentration - time curves are.
References: In the reference list, some of the journal names could be written in a more standardized abbreviated form to be consistent with common scientific citation practices. For example, “Journal of Affective Disorders” could be abbreviated as “J Affect Disord” as is commonly seen in the literature.
Page 2, line 38: “VEN is marketed as an extended and immediate-released dosage form; however, the last mentioned is used far less, because the extended-release dosage form has greater patient adherence, as it is only administered once daily, has a lower risk of ADRs such as nausea and greater therapeutic efficacy [5].” This sentence is a bit convoluted.
Page 3, line 65: “BUP is metabolized in a extensive, stereoselective way in humans” should be “BUP is metabolized in an extensive, stereoselective way in humans.”
Page 5, line 127: “The change in the AUC of VEN, ODV and AM were used to evaluate the clinical relevance of the DDI.” The subject “change” is singular, so the verb should be “was” instead of “were.”
Page 11, line 368: “The percentage of KONBEST data in the fifth to 95th percentile” is an unclear expression. It could be rewritten as “The percentage of KONBEST data that falls within the fifth to 95th percentile.”
Page 14, line 468: “The more sensitive response of (R)-VEN to strong CYP2D6 inhibition compared to (S)-VEN [43] could be a reason for the deviation from the EMA or FDA data.” This sentence structure makes it a bit difficult to understand the relationship between the comparison and the reason for deviation.
Page 17, line 538: “The AUC increase of VEN above 51% calculated by MDDIcalc (based on the AM, as no active metabolites are included in the calculation) differs from the calculation of the MDDI-PBPK model.” This sentence is a bit complex and could be made more straightforward.
Comments on the Quality of English LanguageThe quality of the English language is generally good. The manuscript is written in a clear and understandable manner, with appropriate use of scientific terminology. However, there are a few areas where the language could be improved for greater clarity and precision.
Some sentences are a bit long and complex, which could make them slightly difficult to follow. For example, on page 3, lines 65 - 70: “BUP is metabolized in a extensive, stereoselective way in humans, which results, among others, in the three active metabolites hydroxybupropion (OHB, potency 50% of BUP), threohydrobupropion (THB, potency 20% of BUP) and erythrohydrobupropion (EHB, potency like THB), which are metabolized to inactive metabolites [10,11].”
There are a few instances of incorrect grammar or word usage, such as on page 11, line 368: “The percentage of KONBEST data in the fifth to 95th percentile” should be “The percentage of KONBEST data that falls within the fifth to 95th percentile.”
Additionally, some abbreviations are introduced without always being fully spelled out again in subsequent sections where they might not be as immediately familiar to the reader. For example, after first introducing “TDM” as “therapeutic drug monitoring,” it could be beneficial to briefly remind the reader of the full term when it is used again in a different context later in the paper.
Author Response
This manuscript presents a physiologically based pharmacokinetic (PBPK) model to study the interaction between venlafaxine (VEN) and bupropion (BUP) and its impact on pharmacokinetics in different age groups. The study is relevant as drug-drug interactions are common in pharmacotherapy, especially in treating depression with polypharmacy. The manuscript is generally well-written, with clear objectives, methods, results, and discussion sections. The use of PBPK modeling and comparison with the MDDI calculator is a strength. However, some areas could be improved for better clarity and comprehensiveness.
Thank you for your thoughtful and constructive feedback. We are glad to hear that you found the manuscript well-written and the use of PBPK modeling and the comparison with the MDDI calculator to be strengths. We appreciate your suggestions for improvement and will work on enhancing the clarity and comprehensiveness of the areas you mentioned.
Abstract
Could provide more specific details about the age-relevant changes in the abstract, such as the differences in AUC changes between younger and older patients.
The mention of the need for further research regarding elderly patients, renal or hepatic impairment, and polymorphisms is very brief and could be elaborated on.
Thank you for your insightful comments. You’ve raised excellent points regarding the differences in AUC changes between younger and older patients, as well as the brief mention of the need for further research into elderly patients, renal or hepatic impairment, and polymorphisms. Due to the 250-word limit for the abstract, it was challenging to incorporate all the relevant details. Therefore, we focused on presenting the most concise and impactful findings of the study. However, we fully agree that further elaboration on these aspects would strengthen the discussion. We will be sure to address these points in the full text, ensuring a deeper exploration of age-related differences and other key factors that could influence treatment outcomes. Your feedback is much appreciated!
Introduction
While it mentions the importance of individual pharmacokinetics, it could further discuss how the current study will contribute to personalized medicine in this context.
The introduction could briefly touch on the current state of knowledge regarding the VEN-BUP interaction and what gaps the current study aims to fill.
Thank you very much for your valuable suggestions. We truly appreciate your input. In response to your comment, we have expanded the manuscript to further discuss how the current study contributes to personalized medicine, particularly by highlighting the role of the PBPK model in adjusting drug dosages based on individual patient characteristics. This addition should clarify how the study aims to improve individualized treatment strategies.
Additionally, we have updated the introduction to provide a brief overview of the current state of knowledge regarding the VEN-BUP interaction and to outline the specific gaps that the present study seeks to address. We believe these revisions strengthen the manuscript and better contextualize our research within the broader field. Your feedback has been incredibly helpful in enhancing the clarity and focus of the paper. Thank you again!
Materials and Methods
Some of the technical details, such as the specific equations used in PK-Sim for estimating drug-associated models, could be more clearly presented or referenced for readers who want to understand the modeling process in depth.
Thank you for your valuable feedback. Since the PK-Sim manual, which is publicly accessible, outlines all the specific steps and equations used in estimating drug-associated models, we have explicitly referenced this in the revised manuscript for readers who wish to explore the modeling process in more detail.
The description of the multiple interaction simulation with itraconazole could be more detailed, especially regarding the rationale for choosing this combination and the expected outcomes.
Thank you for your helpful feedback. We have expanded the description of the multiple interaction simulation with itraconazole to provide more detail, including the rationale for selecting this combination and the expected outcomes.
Results
In the DDI model results, while the differences in ctr and AUC between younger and older patients are described, a more in-depth statistical analysis could be performed to determine the significance of these differences.
Thank you for your valuable suggestion. We have now included a statistical analysis to assess the significance of the differences in cmax and AUC between younger and older patients in the supplemental material. Unfortunately, the possibilities in PK-Sim to extract the trough levels of each individual simulation without considerable additional effort are limited, but for AUC and cmax this was possible in terms of time.
The interpretation of the results regarding the clinical relevance of the DDIs could be more explicit, especially in relation to the therapeutic reference range and the potential need for dose adjustment.
Thank you for your constructive feedback. We have made the interpretation of the results regarding the clinical relevance of the DDIs more explicit.
Discussion
Some of the explanations, such as the possible reasons for the deviation from the EMA or FDA data on inhibitory potential, could be further supported by additional references or in vitro studies.
Thank you for your valuable feedback. In response, we have added more supporting information to the explanations, particularly regarding the possible reasons for the deviation from the EMA or FDA data on inhibitory potential. We have now included references to the probe/index substrates used by the FDA and EMA, namely dextromethorphan and desipramine, to provide additional context and clarity. Additionally, we have cited the previously mentioned in vitro study by Fogelman et al., which highlights the involvement of other CYP enzymes in the formation of ODV, further supporting the explanation. We hope this additional information strengthens the discussion, and we greatly appreciate your helpful suggestions!
The discussion on future research directions could be more specific, for example, outlining the exact types of studies needed to address the limitations of the current model and improve the understanding of the VEN-BUP interaction.
Thank you for the suggestion! We’ve now included this point in the manuscript. The discussion on future research directions has been updated to highlight that data collection within the context of Therapeutic Drug Monitoring (TDM) should incorporate renal and liver function parameters. This would enable the extrapolation of the PBPk model to patients with renal and/or hepatic impairment, providing a cost-effective and practical approach to generate relevant data for further evaluation. Your feedback has been very helpful in improving the manuscript!
The discussion on the potential effect of BUP on the enantiomers of VEN could be expanded. Future research directions regarding the stereoselective consideration of the interaction and its pharmacodynamic implications could be more clearly outlined.
Thank you for your thoughtful suggestion. We agree that the discussion on the potential effect of BUP on the enantiomers of VEN could be expanded. In response, we have now included a clearer outline of future research directions, particularly regarding the stereoselective consideration of this interaction. One promising approach for future studies could involve the integration of both pharmacokinetic (PK) and pharmacodynamic (PD) modeling. PK/PD models could provide a deeper understanding of how the stereoselectivity of the VEN-BUP interaction affects therapeutic outcomes and potential side effects. This would be a valuable tool in optimizing the clinical use of these medications, considering their individual enantiomers' distinct pharmacological profiles. Your input has been instrumental in improving the manuscript, and we have incorporated these ideas into the revised version.
Minor issues:
Page 11, Table 4: In the “parameter” column, “CVEN” and “CODV” might be more intuitive if written as “CVEN” and “CODV” to clearly indicate they represent the concentrations of VEN and ODV.
The changes have been made as suggested.
Throughout the manuscript: Some of the abbreviations are introduced and then used without always being fully spelled out again in subsequent sections where they might not be as immediately familiar to the reader. For example, after first introducing “TDM” as “therapeutic drug monitoring,” it could be beneficial to briefly remind the reader of the full term when it is used again in a different context later in the paper.
The changes have been made as suggested.
Figure captions: While the figures are generally helpful, some of the captions could provide more detailed explanations. For instance, in Figure 4, in addition to stating the doses of BUP and the age groups, it could briefly mention what the overall trends or key takeaways from the plasma concentration - time curves are.
Thank you for your helpful suggestion. We appreciate your feedback regarding the figure captions. We have incorporated a more detailed explanation of the overall trends from the plasma concentration-time curves in the main text. This should help improve understanding and provide additional context for Figure 4.
References: In the reference list, some of the journal names could be written in a more standardized abbreviated form to be consistent with common scientific citation practices. For example, “Journal of Affective Disorders” could be abbreviated as “J Affect Disord” as is commonly seen in the literature.
The changes have been made as suggested.
Page 2, line 38: “VEN is marketed as an extended and immediate-released dosage form; however, the last mentioned is used far less, because the extended-release dosage form has greater patient adherence, as it is only administered once daily, has a lower risk of ADRs such as nausea and greater therapeutic efficacy [5].” This sentence is a bit convoluted.
Thank you for your suggestion. The sentence has been split into three parts to improve clarity and readability.
Page 3, line 65: “BUP is metabolized in a extensive, stereoselective way in humans” should be “BUP is metabolized in an extensive, stereoselective way in humans.”
The changes have been made as suggested.
Page 5, line 127: “The change in the AUC of VEN, ODV and AM were used to evaluate the clinical relevance of the DDI.” The subject “change” is singular, so the verb should be “was” instead of “were.”
The changes have been made as suggested.
Page 11, line 368: “The percentage of KONBEST data in the fifth to 95th percentile” is an unclear expression. It could be rewritten as “The percentage of KONBEST data that falls within the fifth to 95th percentile.”
The changes have been made as suggested.
Page 14, line 468: “The more sensitive response of (R)-VEN to strong CYP2D6 inhibition compared to (S)-VEN [43] could be a reason for the deviation from the EMA or FDA data.” This sentence structure makes it a bit difficult to understand the relationship between the comparison and the reason for deviation.
Thank you for pointing that out. We have revised the sentence to improve clarity. The revised version now includes a more detailed explanation of the situation. The updated sentence better clarifies the relationship between the comparison of (R)-VEN and (S)-VEN's sensitivity to strong CYP2D6 inhibition and how this could contribute to the deviation from the EMA or FDA data. We hope this makes the meaning more accessible and the connection clearer. Your feedback was very helpful in improving the readability of this part of the manuscript!
Page 17, line 538: “The AUC increase of VEN above 51% calculated by MDDIcalc (based on the AM, as no active metabolites are included in the calculation) differs from the calculation of the MDDI-PBPK model.” This sentence is a bit complex and could be made more straightforward.
Thank you for your feedback. The sentence has been simplified for better clarity.
Comments on the Quality of English Language
The quality of the English language is generally good. The manuscript is written in a clear and understandable manner, with appropriate use of scientific terminology. However, there are a few areas where the language could be improved for greater clarity and precision.
Thank you for your kind words regarding the quality of the manuscript. We appreciate your constructive feedback, and we will review the areas you mentioned to improve clarity and precision in the language.
Some sentences are a bit long and complex, which could make them slightly difficult to follow. For example, on page 3, lines 65 - 70: “BUP is metabolized in a extensive, stereoselective way in humans, which results, among others, in the three active metabolites hydroxybupropion (OHB, potency 50% of BUP), threohydrobupropion (THB, potency 20% of BUP) and erythrohydrobupropion (EHB, potency like THB), which are metabolized to inactive metabolites [10,11].”
The changes have been made as suggested.
There are a few instances of incorrect grammar or word usage, such as on page 11, line 368: “The percentage of KONBEST data in the fifth to 95th percentile” should be “The percentage of KONBEST data that falls within the fifth to 95th percentile.”
The changes have been made as suggested.
Additionally, some abbreviations are introduced without always being fully spelled out again in subsequent sections where they might not be as immediately familiar to the reader. For example, after first introducing “TDM” as “therapeutic drug monitoring,” it could be beneficial to briefly remind the reader of the full term when it is used again in a different context later in the paper.
The changes have been made as suggested.

Round 2
Reviewer 3 Report
Comments and Suggestions for Authors
The response letter to the comments is comprehensive and shows a positive attitude towards improving the manuscript. The authors have addressed each comment in detail and have provided clear explanations and actions taken to address the concerns raised by the reviewer.
Overall, the revised manuscript has made significant progress and is in a much stronger position than the original one.
